# Feedstock Contract Considerations for a Piedmont Biorefinery

**John Cundiff [1], Robert "Bobby" Grisso [1,*] and John Fike [2]**

[1]   Biological Systems Engineering, Virginia Tech, Blacksburg, VA 24061, USA; jcundiff@vt.edu
[2]   School of Plant and Environmental Sciences, Virginia Tech, Blacksburg, VA 24061, USA; jfike@vt.edu
*   Correspondence: rgrisso@vt.edu; Tel.: +1-540-231-1980

**Abstract:** A biorefinery purchasing feedstock (perennial grass) must offer contracts that provide the same opportunity to earn a profit for a feedstock contractor located 50 or 5 km from the biorefinery. The business plan presented here specifies that the biomass is purchased in satellite storage locations (SSLs), and the load-out and hauling costs are paid by the biorefinery. Contracts can be offered for harvest in September, October, and November, a three-month harvest window, or the harvest window can be extended to December, January, and February, a six-month harvest window. Required total storage capacity is 75% of annual consumption for the three-month window and 50% for the six-month window, a significant difference in total storage capacity (cost). The storage cost difference paid by the biorefinery is 5.27 and 3.52 USD/Mg for the three-month and six-month, respectively. Several issues must be addressed in the feedstock contracts: (1) earlier harvest, before plant senescence, means less nutrients are translocated back into the soil and more are removed at harvest; (2) harvest losses are higher for all harvests after the September harvest; and (3) storage losses increase with storage time in the SSL. Time of removal from the SSL is dictated by the biorefinery; thus, the feedstock contractor must be compensated. The contracts paid by the biorefinery, averaged across the entire annual consumption, were about the same for the three-month window, and six-month window. This result was obtained because fertilizer cost decreases and harvest losses increase as the harvest date increases; thus, the two factors tend to offset. Using a 77 USD/Mg base cost, representative feedstock payment at the SSL (no storage losses included) for contractors with various month contracts are September (84.30), October (85.54), November (86.72), December 88.63), January (89.98), and February (90.58). Subsequent compensation for storage losses depends on the amount of time the particular unit of biomass is in storage before shipment.

**Keywords:** biomass inventory; contracts; harvest windows; switchgrass; feedstocks; storage costs; management systems; satellite storage locations

## 1. Introduction

In order to attract the highest concentration of feedstock produced within a given radius of a potential location, a biorefinery needs to offer feedstock contracts that give each contractor an equal opportunity to earn a profit. These contracts are impacted by the losses that occur when harvest is extended over various months of the harvest season. Other factors include different fertilizer requirements for different harvest dates and losses due to length of storage. This study developed a method for assessing feedstock contracts. It was done by using parameters for a specific location in the Piedmont, a physiographic region encompassing significant parts of five states in the Southeastern USA (VA, NC, SC, GA, and AL).

The study first presents an analysis of three harvest window scenarios and reports the average feedstock cost for annual operation of a biorefinery. Second, the study presents a method for

calculating a feedstock price offered to contractors with contracts to harvest different months over a six-month season.

## 2. Review of Literature

The Southeast (USA) has great potential to produce the biomass quantities needed to supply a developing bioproducts industry [1,2]. Abundant rainfall, long growing seasons, and adapted plant species can support high levels of feedstock production. As well, the region has significant available land that can be converted to this purpose with negligible impact on food production or concern over indirect land use change—itself a concern that may be overblown [3].

We postulate that it is in the national interest to promote development of rural economies [4]. As such, the development of a biorefining industry could play an important role in revitalizing rural Southeast communities that have suffered slower economic growth, and subsequent population declines over the past several decades [5,6].

Rajagopal et al. [7] noted an increased complexity in a farmer's decision to produce feedstocks. Farmers will not produce the feedstock unless a guaranteed market (via contracts) and returns at least as great as alternative enterprises. Similarly, biorefinery investments will be limited unless they can guarantee affordable year-round feedstock supply, among other assurances such as a market for their products, the necessary infrastructure to transport biofuels/bioproducts, and policies which support their industry.

Fewell et al. [8] emphasized that farmers have the ability to produce feedstocks in large volumes but their willingness to do so is unknown. Large-scale biomass production has not been proven to be economically viable, thus uncertainty exists. Farmers' willingness to sign contracts will depend on their knowledge of the agreement and/or required practices and their skills at implementing the requirements [9]. Farmers will grow feedstocks if the returns outweigh production costs, including opportunity costs [7]. However, the production of feedstocks in competition with traditional forages and livestock enterprises will cause prices for these displaced commodities to increase over the long term resulting in increasing competition for dedicated feedstocks [10,11].

Since biomass markets are not established, farmers will need contractual agreements that minimize their risks. Such contracts will establish pricing, harvest parameters (including timeframe), storage requirements, land area requirements, quality levels, and other arrangements between farmers and biorefineries [12–22].

Fewell et al. [8] and Caldas et al. [23] surveyed Kansas farmers and assessed their willingness to grow switchgrass (*Panicum virgatum* L.) under several contract scenarios. Their results showed that contract attributes can affect farmers' potential decisions. Positive attributes included net returns, biorefinery harvest options, insurance availability, and cost share assistance. Contract length negatively impacted farmers' decisions, indicating that farmers prefer not to make long-term commitments to such an uncertain enterprise. From Fewell et al. [8] survey results, they suggested further research is needed on the impact of feedstock characteristics, storage, and transportation issues.

References state the need for a method to calculate a different feedstock price for a different harvest date, and a method to compensate the feedstock contractor for harvest losses in distributed storage. We did not find any reference that gives the needed methods, and offer this study as an initial step to meet this need.

## 3. Objectives

The objectives of this study are as follows: (1) to determine how the different cost factors change under potential feedstock supply scenarios that might be used by a biorefinery in the Southeastern USA, and (2) to develop a method for writing feedstock contracts that incorporates these cost factors in the price offered for feedstock harvested in a given month and shipped after a given storage.

## 4. Materials and Methods

Feedstock density will have primacy in locating biorefining centers. For a given region to be suitable for a biorefinery, the central location must have sufficient feedstock production potential within a given radius. (The reference point used here is a 50 km radius encompassing 7854 km$^2$). This will entail both the feedstock of choice having high per land area output, and significant (>5%) available area adaptable to this purpose. In order to achieve the desired feedstock density, a biorefinery will need to offer a feedstock contract that is attractive to as many farmers as possible. To accomplish this goal, a biorefinery must address two central logistics and supply issues:

1.  All producers within the reference radius must have the same opportunity for profit. A producer located 50 km from the biorefinery must not be disadvantaged relative to a producer located 5 km from the biorefinery. We argue that the highway hauling cost must be borne by the biorefinery, thus transportation is not part of the feedstock contract.
2.  The feedstock contract must compensate for storage losses. Significant storage will be required to meet a biorefinery's supply needs. Distributed, satellite storage locations (SSLs), stationed within a defined distance to the production field, would benefit the biorefinery by facilitating biomass collection and delivery scheduling. However, the SSLs will need to be filled and emptied at various times over the course of a year, and a producer whose feedstock is stored 6 months before shipment will incur a higher storage loss than a producer whose biomass is shipped shortly after harvest.

Another key, although perhaps underappreciated issue, is the effect of the "harvest window". Length of the harvest season (the harvest window) for an herbaceous feedstock significantly impacts average delivered cost of that material for year-round biorefinery operations. This occurs because of the harvest window's effect on the required maximum satellite storage capacity, which is a critical cost factor for any biorefinery using an herbaceous feedstock.

To ease logistics constraints and storage demands, a biorefinery would like to offer feedstock contracts that provide the lowest average delivered cost of feedstock, and the harvest window impacts this value. Storage and logistics costs can be reduced if the biorefinery has the ability to "campaign" feedstocks, i.e., the facility processes one feedstock for certain months then switches the operations to utilize other feedstocks for the remainder of the year.

For the biorefinery, it is certainly preferable to receive directly harvested material for 6 months and only need storage capacity for a maximum of 6 months' supply. Contrast this with a feedstock such as corn stover, in which the harvest window is not more than about one month and storage is required for about an 11-month supply for feedstock. However, an expanded harvest window may have several agronomic and economic implications on system productivity and resulting profitability to the feedstock producer—which must be dealt with in the feedstock contract.

Our intent is to determine how the different cost factors change under potential feedstock supply scenarios that might be used by a biorefinery in the Southeastern USA. Here, we present analyses of three scenarios for seasonal herbaceous biomass harvest and year-round delivery. These include the following:

1.  Harvest over three months, with storage and delivery for year-round operation.
2.  Harvest over six months, with storage and delivery for year-round operation.
3.  Harvest over six months, with a relatively short (maximum of 3 months) storage and delivery for 6-month operation. The remaining 6 months of biorefining operations would be supplied with another feedstock. (In the Southeast, this may be some type of woody feedstock).

The scenarios were compared in terms of the average delivered cost of the herbaceous feedstock only; logistics for woody feedstock delivery was not part of this study.

The business model for this study presumes that the biorefinery will contract with feedstock producers, or perhaps a broker representing a group of smaller producers, to grow, harvest, and place

feedstock in SSLs. The contract holder will build and maintain the SSL, a graded and graveled surface for single layer ambient storage of round bales, and be paid a storage fee as part of their contract. Hauling from the SSLs to meet weekly demand will be scheduled by the biorefinery. The details of this scheduling (when an SSL is emptied and refilled) is not covered in this study; such work has been highlighted by others [24–45].

Switchgrass was used as the herbaceous feedstock of choice for this study. Switchgrass, a warm-season perennial grass, has proven highly productive over a wide range of climatic and edaphic conditions, and is native to much of the continental USA [46]. The crop can be harvested and stored with haying equipment, making it compatible with many existing farming operations in the region [47,48].

The example siting for the biorefinery was in the Piedmont physiographic region. The Piedmont covers a significant part of five Southeastern States (Alabama, Georgia, North Carolina, South Carolina, and Virginia), and is characterized by small- and intermediate-size farms. The analysis was performed considering production occurred in Virginia, west of Lynchburg (Virginia, USA).

Land available in the region generally is not used for grain or other commodity crops. More typically, farmland is devoted to pasture and hay production. As such, an energy crop such as switchgrass is both compatible with, or suitable as a replacement for, the existing forage-livestock enterprises.

Estimates of switchgrass production on the region's dominant soil types (ultisols) were based on results from field-scale research on a representative site [49], using a moderate fertility regime to meet nitrogen replacement. Feedstock nitrogen concentrations and consequent input needs were approximated based on data from Fike et al. [50], who reported nitrogen concentrations for "Alamo" switchgrass were about 13 g/kg in July and 5 g/kg in November. We assumed a monthly decrease in switchgrass nitrogen of 2 g/kg, which is supported by other studies from the region in which biomass nitrogen concentrations declined with delayed harvest [51–53].

Harvest calculations were based on the use of round bales. The round bale was selected for the analysis because it is the predominant hay harvest machine used in the Piedmont region. The assumption is that farmers entering into feedstock production contracts would use their existing production equipment, which they currently operate for their livestock enterprise. Increasing the equipment's annual operating hours reduces USD/h operating cost. Following harvest, farmers will move bales to the SSL, placing them in single-layer ambient storage. Load-out and highway hauling are the responsibility of the biorefinery.

## 4.1. Length of Harvest Window Cost Factors

The four harvest-window factors with the greatest impact on the average delivered cost of an herbaceous feedstock are as follows:

1. Size/Cost of Storage—e.g., does storage have to be provided for an 11 month-supply (1-month harvest window) or a 6-month supply (6-month harvest window)?
2. Biomass Losses—in-field losses incurred because of delayed harvest, machinery-related harvest losses, and losses in transport and storage.
3. Fertilizer Cost—greater fertilizer costs for stand maintenance occur if the crop is harvested before senescence, because nutrient translocation into the root system is reduced, which increases removal of nutrients (which must be replaced).
4. Harvest Cost—higher annual use hours of harvest equipment reduce USD/h ownership cost, and thus reduces USD/Mg harvest cost.

## 4.2. Analysis for Comparison of Three Harvest Window Scenarios

The analysis herein compares the three harvest-window scenarios from industry—feedstock contactors and biorefinery—perspectives. A later section focuses specifically on the contract offered to individual feedstock contractors.

In Virginia's Piedmont region, peak standing crop likely occurs in September, but gathering all available biomass in a narrow window of time presents significant cost and logistics challenges. Thus, a feedstock supply system based on perennial grasses is likely to occur over a much longer time frame, with delayed harvest possible even into March. However, an extended harvest season presents systemic tradeoffs, as the ease for supply logistics is met with lower yield due to in-field weathering losses (e.g., leaf shatter) and harvest-related losses due to machine factors (e.g., shattering during field operations and incomplete biomass pickup).

We tested three different harvest scenarios to explore this "yield vs. logistics" dynamic. The first scenario tests system costs over a 3-month harvest window: September through November. The second scenario extends the harvest window to six months: September through February. Although the number of months is doubled, the number of possible workday hours for harvest operations is not, because weather conditions during the winter months limit the percentage of the total biomass that can be harvested during December, January, and February. Expected workday hours for Lynchburg, VA, USA (Table 1) are taken from Grisso and Webb [54]. These workday hours were used for the simulations in this study. The percentage of the annual total (719 h) in each month is also given in Table 1. It is interesting that less than 27% of the total harvest occurs in the last half of the 6-month window. The total for a 3-month season is 527 h.

**Table 1.** Expected monthly workday hours in Lynchburg, VA, USA [54].

| Harvest Month | Probable Workday Hours (Total) | Scenario 1 (% Annual Total) | Scenarios 2 and 3 (% Annual Total) |
|---|---|---|---|
| 1 (September) | 196 | 37.2 | 27.3 |
| 2 (October) | 185 | 35.1 | 25.7 |
| 3 (November) | 146 | 27.7 | 20.3 |
| 4 (December) | 66 | | 9.1 |
| 5 (January) | 63 | | 8.8 |
| 6 (February) | 63 | | 8.8 |

4.2.1. Definition of Parameters

1.  Biorefinery design capacity: Simulations were preformed based on plant capacities of 0.5, 1, and 2 bale/min for a biorefinery operating 24/7, 48 wk/y. (Results are given on the basis of USD/Mg-annual-capacity; thus, biorefinery size does not impact the interpretation of results).

2.  The round baler will create 5 × 4 bales {5 ft diameter × 4 ft wide (1.5 m × 1.2 m) round bales}. Average bale weight is 400 kg at 15% MC (wb). As a reference point, the annual biorefinery capacity consuming 1 bale/min for 24/7 operation, 48 wk/y is {400 (60) (24) (7) (48)}/1000 = 193,536 Mg.

3.  Average productivity across all balers employed for the harvest, and over all hours these balers work, is taken to be 8.2 Mg/h. This represents a conservative estimate for the region, and accounts for available production areas, which typically comprise small irregular-shaped fields over rolling terrain. Thus, field size, shape, and topography reduce equipment productivity, and operating time is lost when the balers must travel between multiple relatively small fields.

4.  Average base yield across the entire production area is 6.7 Mg/ha. Although yields greater than 13.4 Mg/ha have been obtained in research plots [55], such estimates typically do not provide an accurate estimate of yields achieved with field-scale equipment. As well, biomass feedstock operations likely will compete with existing uses of marginal land and the poorest quality fields. Previous, regionally appropriate research under such conditions [50] suggests 6.7 Mg/ha is a sufficiently conservative yield estimate. On that basis, the production area required to supply a bale-a-min biorefinery is 28,790 ha, or 3.7% of the land area within a 50-km radius.

5.  Average maintenance fertilizer cost is defined for this study by using an estimate of 7 g of nitrogen per kg of biomass. If nitrogen cost is 1.54 USD/kg, then this is about 72 USD/ha cost to replace harvested nitrogen and does not include cost of other nutrients.

6. Biomass in round bales, placed in single-layer ambient storage at an SSL, is valued at a baseline cost of 77 USD/Mg. This estimate is used for calculating lost revenue adjustments for harvest and storage losses.

7. The SSL is a graded and graveled surface. In this study, the cost to own and maintain an SSL is assigned to be 1.47 USD/m² (see Appendix A).

### 4.2.2. Baler Cost

In this analysis, baler cost is used as a surrogate, or representative, for harvest machinery cost. There may be 300, or more, feedstock producers growing switchgrass for the hypothesized biorefinery. (As previously mentioned, the chosen study area in the Piedmont of Virginia is characterized by small- and intermediate-sized farms, thus there could be many relatively small feedstock producers). It is unlikely that a biorefinery will desire to contract with 300+ individual feedstock producers. We present the following discussion to provide some context.

The cost (USD/h) to operate a baler (tractor cost and labor cost not included) was calculated by using the procedures given in ASABE [56], and these data were used to plot Figure 1. A baler needs to operate about 200 h/y to reach the ownership cost curve asymptote.

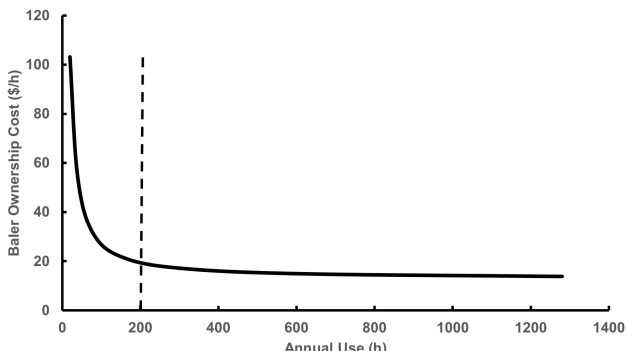

**Figure 1.** Baler operating cost ($/h) as a function of annual use hours.

With our assumption of average baler performance of 8.2 Mg/h and average yield of 6.7 Mg/ha, the field capacity is 0.82 ha/h. A feedstock contractor, in order to achieve a favorable baler cost, would like a production unit of 200 h/y × 0.82 ha/h = 164 ha. If a farmer uses their baler 50 h for another enterprise on their farm (hay harvest), then they would like to have a contract for 150 h/y × 0.82 ha/h = 123 ha. A 123-ha contract is relatively large given the size of the farming operations in the five Piedmont states. What business structure is needed for a feedstock logistics chain that will provide opportunity for small- and intermediate-size farmers?

We postulate that there will be a range of different business plans that might emerge. Examples are as follows:

1. Certain farmers, or full-time contract harvesters, will enter agreements with their neighbors to harvest their biomass and accumulate an economic production unit. These farmers will own and operate one or more SSLs.

2. Brokers will contract for farmers to harvest (using their own equipment) and deliver biomass to an SSL owned and operated by the broker.

For this analysis, we assume a distribution of feedstock contracts given in Table 2. With an average yield of 6.7 Mg/ha (no accounting for harvest losses at this point), the total area harvested to supply the biorefinery's annual consumption is 28,790 ha. The number of contracts is 263 with an average size of 109 ha.

**Table 2.** Assumed distribution of feedstock contracts.

| Percentage of Contracts | Size of Contract (ha) | Number of Contracts | Area (ha) | Percentage of Total Area |
|---|---|---|---|---|
| 41 | 40 | 108 | 4320 | 15 |
| 42 | 123 | 110 | 13,530 | 47 |
| 17 | 243 | 45 | 10,940 | 38 |
| 100 | | 263 | 28,790 | 100 |

Here we assume the following: (1) all balers have an average field capacity of (6.7 Mg/ha) / (8.2 Mg/h) = 0.82 ha/h, and (2) all balers are used 50 h for an enterprise other than the feedstock contract. A vendor with a 40- ha contract will operate their baler 40 ha/0.82 Mg/h = 48.8 ~ 49 h, for a total of 99 annual operating hours (49 + 50 = 99 h). In like manner, a vendor with a 123 ha contract will use their baler 210 h/y, and a vendor with a 243 ha contract will use their baler 346 h/y. Referencing the curve in Figure 1, these annual uses give the following baler per-hour operating costs: 40 ha (27.26 USD/h), 123 ha (20.66 USD/h), and 243 ha (16.53 USD/h). Using the percentages of land area in Table 2, the weighted average per-hour baler operating cost for a fleet of balers to harvest the production is given by Equation (1):

$$0.15(27.26) \ + \ 0.47(19.48) \ + \ 0.38\,(16.53) \ = \ 19.53 \text{ USD/h} \tag{1}$$

Using the average baler capacity, this corresponds to (19.53 USD/h) / (8.2 Mg/h) = 2.38 USD/Mg. The weighted annual operating hours for a "fleet" baler can be computed as follows:

$$0.15(99) \ + \ 0.47(200) \ + \ 0.38(346) \ = \ 240 \text{ h} \tag{2}$$

This corresponds to 45.5% of the total available hours (527 h) in the 3-month window and 33% of the hours (719 h) in the 6-month scenario.

If, averaged across the fleet, the balers utilize 70% of the total available workday hours (369 h in the 3-month window, 503 h in the 6-month window), then Scenario 1 would require a minimum of 65 balers, versus a minimum 48 balers for Scenario 2. Since half the total production area is harvested for Scenario 3, the required minimum number of balers is 24 units.

A minimum baling cost could be achieved if 65 contractors could operate 65 balers and harvest the entire production area in 3 months. Considering the distribution of production fields across the 50 km radius, and several hundred different owners of these production fields, this will never be a practical option. In fact, it is undesirable, even if it could be achieved.

The harvest operation is the most weather-dependent operation in the feedstock logistics chain for a biorefinery. The supply chain must be organized to take advantage of each suitable harvest day. It is in the biorefinery's interest to have more, rather than fewer, balers ready to deploy when weather and field conditions are suitable. A greater number of balers (number of contracts) means fewer annual operating hours per baler, with corresponding greater operating cost (USD/h). We postulate that the resulting small increase in baling cost is a very cost-effective investment by the industry (feedstock contractors and biorefinery). We consider a harvest plan that is designed for adaptation to variable weather in the harvest window to be a robust plan. A plan with 200 to 300 vendors is much more robust than a plan with the 65 minimum units.

For this analysis, we use the following assumption. Average operating hours for a baler operating in the 3-month window is 527/719 = 0.73 times the average weighted operating hours given in Equation (2), 0.73 × 240 = 175.2 ~ 175 h.

The baling cost (Figure 1) corresponding to 175 h in Scenario 1 and 240 h in Scenarios 2 and 3 is as follows:

Scenario 1 = 2.51 USD/Mg;
Scenarios 2 and 3 = 2.24 USD/Mg.

### 4.2.3. Fertilizer Cost

Area harvested each month is calculated with Equation (3). These areas are used in the total fertilizer cost calculation.

$$\text{har}_i = \frac{D_c}{Y_d(1 - HL_i)} \, \text{pwd}_i \tag{3}$$

where $\text{har}_i$ = harvested area (*i*th month) (ha); $D_c$ = biorefinery design capacity (Mg/y); $Y_d$ = base yield (at September harvest) across feedstock production area (Mg/ha); $\text{pwd}_i$ = probable workday hours *i*th month, fraction for Scenario 1, 2, or 3 (dec); and $HL_i$ = harvest loss *i*th month (dec).

Fertilizer costs for two production sites (identified as high- and low-productivity fields) are given in Figure 2. The total area harvested each month, after accounting for harvest losses, is multiplied by the fertilizer value of the nitrogen removed with a harvest in that month (Figure 2). This total fertilizer cost is divided by biorefinery design capacity to obtain an average fertilizer cost (USD/Mg) for comparison of the three scenarios.

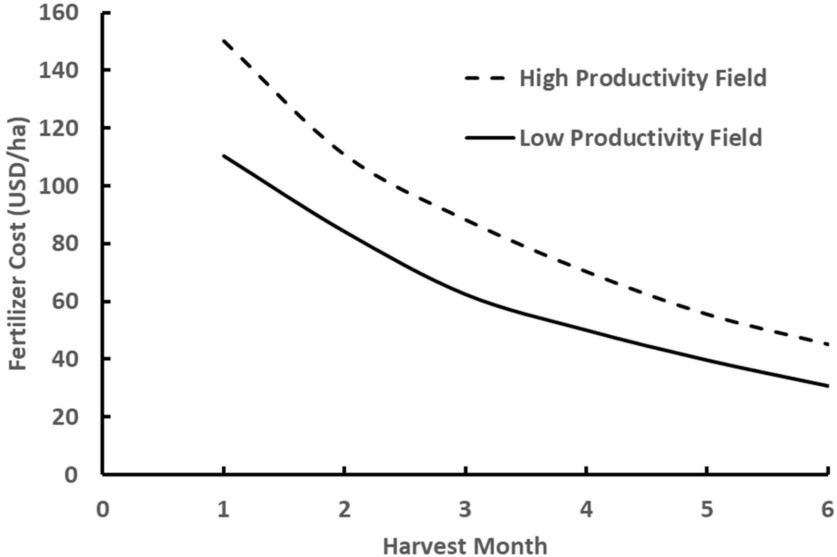

**Figure 2.** Nitrogen fertilizer replacement cost for biomass harvested under high- and low-productivity site conditions in the Virginia Piedmont when switchgrass is harvested from (1) September through (6) February.

### 4.2.4. Harvest Losses

The biomass harvested the *i*th month is multiplied by the harvest loss factor (obtained from Figure 3) to give the estimated harvest losses associated with the area harvested that month. These harvest losses are multiplied by the "value" of the biomass at the time of harvest {$V_b + V_{HLi}$ (harvest loss adjustment for *i*th month—Equation (5))} USD/Mg to get the total harvest loss cost for a given scenario. As was done with fertilizer, this total cost is divided by biorefinery design capacity to get a USD/Mg cost for comparison of the three scenarios. Note, the biorefinery capacity for Scenario 3 is half that for Scenarios 1 and 2.

### 4.2.5. Storage Losses

Cundiff and Marsh [57] reported the losses measured for single layer ambient storage of round bales of switchgrass on a crushed rock surface. In their study, care was used to remove all material from storage, thus eliminating handling loss from the measurement. Storage losses ranged from near zero to 5% over a 6-month period. Mooney et al. [58] fit a quadratic plateau model to their experimental data to describe the loss curve for switchgrass round bales stacked in a single layer on a crushed rock

surface and held under ambient conditions. Their model predicted an increase in storage loss with time up to some plateau. The amount of this monthly increase got smaller as the months in storage increase. The same shaped curve was used for this study. As shown in Figure 4, the maximum values used were 5.0% and 8.5% for the low- and high-loss curves, respectively.

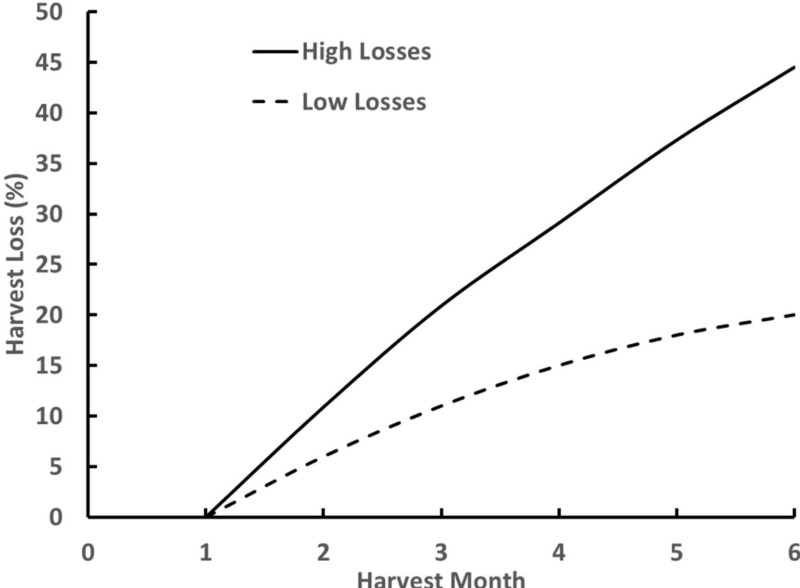

**Figure 3.** Harvest loss factors as a function of harvest date. Harvest months are (1) September to (6) February.

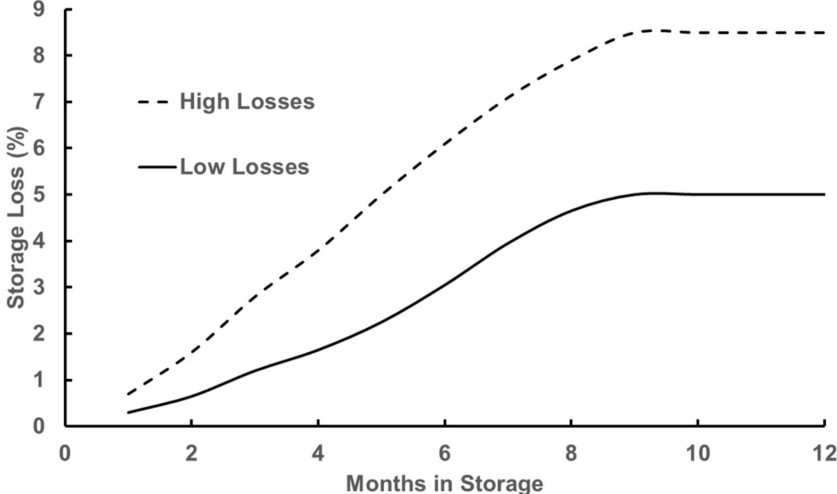

**Figure 4.** Storage loss factors for round bales of switchgrass in single-layer ambient storage on a gravel surface.

The amount harvested (harvest losses accounted for) each month across the entire harvest region, minus the biorefinery consumption (direct delivery) for that month, gives the total amount placed in storage for that month. Our procedure assumes a first-in, first-out rule for the delivery of stored biomass from an SSL. Using Scenario 1 as an example, the material stored in September is used to meet demand in December, January, and so forth, until it is completely consumed. Then, the material stored in October is shipped, followed by the material placed in storage in November.

The analysis was performed by using 4033 Mg—the feedstock consumed in a week at the biorefinery—as the process unit. A bale-per-min processing plan will consume 10,080 bales/wk. If a truckload holds 40 bales [59], and deliveries are received 6 d/wk, the average per-day delivery needs to be at least 42 loads. For each process unit withdrawn from an SSL, the mass of the unit (Mg) times the loss factor for the number of months it was held in storage was used to calculate a storage loss for that unit. These losses were summed for each process unit delivered during the year to give a total storage loss for annual operation.

The total loss times the biomass value when it was placed in the SSL $\{V_b + V_{HLi}$ (Equation (5))$\}$ (USD/Mg) gives the total cost of the storage loss. This total storage loss cost is divided by plant design capacity (halved for Scenario 3) to give a USD/Mg for comparison of the three scenarios.

The Scenario 3 analysis requires further comment. The entire harvest in December was not sufficient to meet that month's demand, and the same was true of January and February. Thus, stored material needs to be delivered to make up the harvest deficit for each of these months. Consequently, the weekly shipment from storage varied week to week. Results were verified within the simulation program to ensure the monthly demand was delivered each month.

### 4.3. Computation Procedure

Three MATLAB programs were written, one for each scenario. These MATLAB programs are identified as Scenario 1 MATLAB program, Scenario 2 MATLAB program, and Scenario 3 MATLAB program (these programs can be accessed in the Supplementary Materials). These programs have extensive comments to explain the computation procedure, and they are available for a user that wants a line-by-line explanation of how the results were obtained.

For Scenario 1 and Scenario 2, the needed biorefinery demand is delivered directly each month. Amounts placed in storage in months 4–6 (Scenario 2) were very small. (In fact, the total stored during these three months was not quite enough to meet one week's demand at the biorefinery). For Scenario 3, the direct-shipped biomass from the December, January, and February harvests was not sufficient to meet demand, and additional biomass had to be supplemented from stored inventory.

Different weather conditions during the weeks of storage were not considered. For example, some biomass harvested the first week in September was shipped the first week in Mar. It was exposed to fall and winter weather conditions. Some biomass harvested the first week in November was shipped in late spring, thus it was exposed to winter and early spring weather conditions. If both these materials were stored the same number of weeks, the same storage loss factor from the storage loss curve—corresponding to that number of weeks in storage—was used.

The three scenarios were compared by computing a "total" cost as follows:

Storage Cost + Losses Cost (Harvest + Storage) + Fertilizer Cost + Baler Cost = Total

### 4.4. Considerations for Individual Feedstock Contracts

A feedstock contractor (individual farmer or broker) might obtain a September harvest contract, meaning that they are expected to harvest a specified area in September. Similar contract specifications would apply to other months of harvest. A contractor could get a September/February contract to harvest some of their production area and fill the SSL in those months. A larger broker might have contracts to harvest in all six harvest months.

Direct-shipped biomass will need to be delivered to an SSL where it would be aggregated and shipped to the biorefinery within a few days. This will free SSL space for biomass storage after all deliveries of direct-harvested biomass are completed.

We hypothesize that contracts will need to be adjusted based on the harvest date.

1.  Fertilizer adjustment—A fertilizer adjustment was defined with the cost for a November ($i = 3$) harvest as the base. This means that a September and October harvest will receive a positive adjustment, a November harvest a zero adjustment, and December, January, and February harvests receive a negative adjustment.

$$F_{afi} = F_{fi} - F_{f3} \tag{4}$$

where $F_{afi}$ = fertilizer adjustment factor for $i$th month (USD/ha), $F_{fi}$ = fertilizer cost for biomass harvested in $i$th month (USD/ha), and $F_{f3}$ = fertilizer cost for biomass harvested in November ($i = 3$) month (USD/ha).

These adjustments were applied to the area harvested each month to obtain a total cost of the fertilizer adjustment for that month. (Note, area harvested each month is increased to account for harvest loss). This total is divided by the total Mg harvested that month to obtain a USD/Mg cost for the fertilizer adjustment in the contract offered for that month.

2.  The adjustment paid to growers who harvest after September (and thus incur a harvest loss) is calculated as follows:

$$V_{HLi} = \frac{HL_i}{(1 - HL_i)} V_b \tag{5}$$

where $V_{HLi}$ = payment to compensate for harvest loss in $i$th month (USD/Mg), $HL_i$ = harvest loss in $i$th month (dec), and $V_b$ = base payment for biomass (USD/Mg).

For example, if the harvest losses for the $i$th month are 10%, $HL_i = 0.1$, the adjustment paid is as follows:

$$V_{HLi} = \frac{0.1}{(1 - 0.1)} 77 = 8.56 \frac{USD}{Mg}$$

The payment for biomass harvested and placed in SSLs in the $i$th month is $V_b + V_{HLi}$ = 77 + 8.56 = 85.56 USD/Mg. If the base yield (6.7 Mg/ha) is harvested, and the base price is paid (77 USD/Mg), the gross income is 6.7 (77) = 515.90 USD/ha. If the yield is reduced by a 10% harvest loss, 6.7 (1 − 0.1) = 6.03 Mg/ha, and the payment adjusted as shown, the gross income is 6.03 (85.56) = 515.90 USD/ha, and the feedstock contractor loses no income because of harvest loss.

The harvest loss adjustment for a September harvest is zero. For the other months (low harvest loss curve), the adjustments for Scenario 2 are 4.93 (October), 9.54 (November), 13.62 (December), 16.94 (January), and 19.29 (February). The total biomass harvested in a given month is multiplied by the adjustment to obtain the "cost" of the harvest loss for that month.

Storage losses represent biomass that the biorefinery has paid the feedstock contract holder to store in an SSL, but then does not receive on the day it is delivered. The total losses (Mg) for the $i$th month is multiplied by the value ($V_b + V_{HLi}$) owed to the feedstock contractor when they place the biomass in the SSL, and these costs are summed over all harvest months to obtain a total cost of storage losses.

Total annual cost of feedstock, paid to the feedstock contractors for biomass stored in SSLs, plus the cost of losses during storage, is calculated as follows:

$$TC = D_c V_b + TC_f + TC_{HL} + TC_{SL} \tag{6}$$

where TC = total cost of feedstock to biorefinery (USD), $D_c$ = biorefinery design capacity (Mg), $V_b$ = base payment (USD/Mg), $TC_f$ = total cost fertilizer adjustment (USD), $TC_{HL}$ = total cost harvest loss adjustment (USD), and $TC_{SL}$ = total cost of storage loss (USD).

Total biomass delivered to the biorefinery is the design biorefinery capacity (biomass harvested and delivered direct, or placed in SSLs) minus the total storage losses. This biomass is an "achieved capacity". The average delivered USD/Mg achieved capacity is then as follows:

$$TC_a = \frac{TC}{D_c - SL} \tag{7}$$

where $TC_a$ = achieved feedstock cost (USD/Mg), TC = total cost of feedstock to biorefinery (USD), $D_c$ = annual design capacity (Mg), and SL = total storage losses (Mg).

A storage loss factor was calculated for each harvest month. The total storage loss (Mg) for all units shipped from the $i$th month storage was summed to get a total loss for all these units. This total was divided by the total (Mg) placed in storage that month. The factor is defined for the $i$th month, as follows:

$$SL_{fi} = \frac{SL_i}{S_i} \tag{8}$$

where $SL_{fi}$ = storage loss factor for $i$th month (Mg/Mg), $SL_i$ = total storage loss for all biomass shipped from the $i$th month storage (Mg), and $S_i$ = total biomass stored in $i$th month (Mg).

Each unit shipped from an $i$th month harvest has a different storage loss because each is stored a different number of weeks. It is not practical to calculate a separate rate factor for each unit. The procedure in Equation (8) gives a "representative" factor for all the units shipped from the $i$th month storage.

Typical Feedstock Contracts with Adjustments

This section presents a methodology for calculation of the contract price paid to individual contract holders with a given harvest month contract. Contract holders would like to see a USD/ha payment for comparison of the biomass enterprise with gross income from other enterprises on their farm. Therefore, the harvest loss and fertilizer adjustments are applied as shown in Equation (9) to obtain a USD/ha payment for an $i$th month contract.

$$FCha_i = (V_b + V_{HLi}) \, Y_b \, (1 - HL_i) + F_{afi} \tag{9}$$

where $FCha_i$ = feedstock payment for $i$th month contract (USD/ha), $V_b$ = base payment (USD/Mg), $V_{HLi}$ = harvest loss adjustment for $i$th month contract (Equation (5)) (USD/Mg), $Y_b$ = base yield (Mg/ha), $HL_i$ = harvest loss for $i$th month (dec), and $F_{afi}$ = fertilizer adjustment for $i$th month contract (Equation (4)) (USD/ha).

The payment to a feedstock contractor (USD/Mg) for biomass harvested the $i$th month and delivered direct (direct harvest) is calculated as follows:

$$FCMgd_i = \frac{FCha_i}{Y_b \, (1 - HL_i)} \tag{10}$$

where $FCMgd_i$ = feedstock payment for direct delivery, $i$th month contract (USD/Mg); and $FCha_i$ = feedstock payment for $i$th month contract (Equation (9)) (USD/ha).

When biomass is *delivered* from storage, an adjustment for storage losses is applied. The feedstock contract holder does not control the time in storage; this is controlled by the biorefinery. Thus, an adjustment for storage losses is owed to the contract holder so their opportunity for profit is equal to all other contract holders. This adjustment is given by the following:

$$SL_{afi} = SL_{fi} \, (V_b + V_{HLi}) \tag{11}$$

where $SL_{afi}$ = storage loss adjustment for $i$th month (USD/Mg placed in storage), $SL_{fi}$ = storage loss factor for $i$th month [Equation (8)] (Mg loss/Mg placed in storage), $V_b$ = base payment (USD/Mg), and $V_{HLi}$ = harvest loss adjustment for $i$th month (Equation (5)) (USD/Mg).

The payment for biomass delivered from storage, with storage loss adjustment, is as follows:

$$FCMgs_i = \frac{FCha_i}{Y_b \ (1 - \ HL_i)} + \ SL_{afi} \tag{12}$$

where FCMgsi = feedstock payment for delivery from storage, $i$th month contract (USD/Mg); and SLafi = storage loss adjustment for $i$th month (Equation (11)) (USD/Mg in storage).

## 5. Results and Discussion

As previously stated, all cost factors calculated for comparison of the three scenarios are reported as USD/Mg of design biorefinery capacity. The analysis was performed for biorefineries consuming 0.5, 1.0, and 2.0 bales/min to verify that plant size is not a factor in the results. The key result is the *difference* in the total cost for the various harvest strategies (Scenarios).

The feedstock density needed to supply a 1-bale/min biorefinery is calculated by dividing the required total production of 193,536 Mg by 7854 km², the area within a 50- km radius. The result is an average distribution of 24.64 Mg/km².

Accounting for harvest losses (the low-loss curve), the average yield across all areas harvested for Scenario 2 is 6.12 Mg/ha, which compares favorably to 6.7 Mg/ha, the average yield estimate across all areas when harvests occur in September. Results for all three scenarios are given in Table 3.

**Table 3.** Required production area (accounting for harvest loss) and achieved average yield across entire production area. (Results obtained with Scenarios 1, 2, and 3 MATLAB programs).

| Harvest Window | Harvest Loss | Required Production Area (ha) | Percent in 50- km Radius | Achieved Average Yield (Mg/ha) |
|---|---|---|---|---|
| 3-month (Scenario 1) | Low | 30,421 | 3.9 | 6.36 |
| | High | 32,134 | 4.1 | 6.02 |
| 6-month (Scenario 2) | Low | 31,636 | 4.0 | 6.12 |
| | High | 35,849 | 4.6 | 5.40 |
| 6-month (Scenario 3) | Low | 15,818 | 2.0 | 6.12 |
| | High | 17,925 | 2.3 | 5.40 |

### 5.1. Inventory

The biomass harvested and placed in storage is given in Table 4a (Scenario 1), Table 4b (Scenario 2), and Table 4c (Scenario 3). The inventory for September-, October-, and November-harvested biomass plus the total inventory in SSLs is given in Figure 5 for Scenario 1, Figure 6 for Scenario 2, and Figure 7 for Scenario 3. Note that the total in storage for Scenario 2 continues to increase for weeks 13 through 24 due to the small amount placed in storage, after the direct delivery requirement is satisfied, from the December, January, and February harvests.

Using the first-in, first-out rule, the September material is shipped and then the October and November material. For example, in Scenario 2, shipment of the September material begins in week 25 after the direct delivery from the February harvest is completed. Shipment of the October material begins week 34 and the November material week 42.

The maximum storage requirement for Scenario 1 is 153,245 Mg (79% of biorefinery design capacity) as compared to 104,850 Mg (54%) for Scenario 2. The maximum for Scenario 3 is only 24,496 Mg (25% of 96,786 Mg annual requirement for this scenario). The percentages in Scenarios 1 and 2 are higher than an expected 75% and 50%, respectively, because an extra storage unit (one week's

supply) is stored. In commercial practice, there will always be some number of weeks of extra storage. This issue is not dealt with here.

**Table 4.** (**a**) Biomass harvested, delivered direct, and placed in storage (Scenario 1) (results obtained with Scenario 1 MATLAB program). (**b**) Biomass harvested, delivered direct, and placed in storage, storage (Scenario 2) (results obtained with Scenario 2 MATLAB program). (**c**) Biomass harvested, delivered direct, and placed in storage, storage (Scenario 3) (results obtained with Scenario 3 MATLAB program).

| | | (a) | |
|---|---|---|---|
| **Harvest** | **Shipped Direct** | **Placed in Storage** | **Total** |
| **Month** | **(Mg)** | **(Mg)** | **(Mg)** |
| 1 (September) | 16,131 | 55,862 | 71,993 |
| 2 (October) | 16,131 | 51,821 | 67,952 |
| 3 (November) | 16,131 | 37,496 | 53,627 |
| **Total** | **48,393** | **145,179** | **193,572** |

| | | (b) | |
|---|---|---|---|
| **Harvest** | **Shipped Direct** | **Placed in Storage** | **Total** |
| **Month** | **(Mg)** | **(Mg)** | **(Mg)** |
| 1 (September) | 16,131 | 36,637 | 52,768 |
| 2 (October) | 16,131 | 33,676 | 49,807 |
| 3 (November) | 16,131 | 23,176 | 39,307 |
| 4 (December) | 16,131 | 1638 | 17,769 |
| 5 (January) | 16,131 | 830 | 16,961 |
| 6 (February) | 16,131 | 830 | 16,961 |
| **Total** | **96,786** | **96,786** | **193,572** |

| | | (c) | |
|---|---|---|---|
| **Harvest** | **Shipped Direct** | **Placed in Storage** | **Total** |
| **Month** | **(Mg)** | **(Mg)** | **(Mg)** |
| 1 (September) | 16,131 | 10,255 | 26,386 |
| 2 (October) | 16,131 | 8772 | 24,903 |
| 3 (November) | 16,131 | 3522 | 19,653 |
| 4 (December) | 8884 | 0 | 8884 |
| 5 (January) | 8481 | 0 | 8481 |
| 6 (February) | 8481 | 0 | 8481 |
| **Total** | **74,239** | **22,550** | **96,788** |

*5.2. Storage Cost*

Scenario 1 requires a total storage area (sum of all SSL areas) of 69.5 ha as compared to 46.3 ha for Scenario 2. The lowest storage capacity is required for the six-month campaign (Scenario 3), 10.8 ha. Costs, USD/Mg biorefinery design capacity, are given below:

Three-month harvest: 5.27 USD/Mg;
Six-month harvest: 3.52 USD/Mg;
Six-month harvest, six-month campaign: 1.64 USD/Mg.

*5.3. Harvest and Storage Losses*

It is re-emphasized, the storage losses presented in this section are the total losses obtained by summing the calculated loss for each storage unit delivered for annual biorefinery operation. The calculated losses, harvest and storage, for the three scenarios are given in the following tables: three-month harvest window (Scenario 1) in Table 5a, six-month harvest window (Scenario 2) in Table 5b,

and six-month window, six-month campaign (Scenario 3) in Table 5c. As expected, the harvest losses are greater for Scenarios 2 and 3 since more biomass is harvested after leaf shatter and other losses that occur due to weathering as the switchgrass stands in the field. Moreover, logically, the storage losses are greater for Scenario 1 since more material is stored, much of it for a longer period.

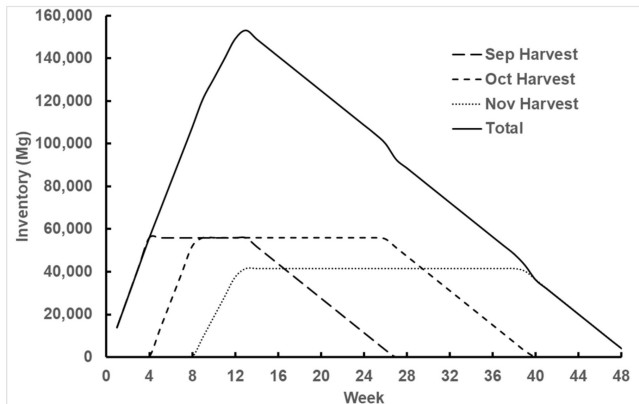

**Figure 5.** Inventory in satellite storage locations (SSLs) for weeks 1 through 48 for the three-month window. (Results obtained with Scenario 1 MATLAB program).

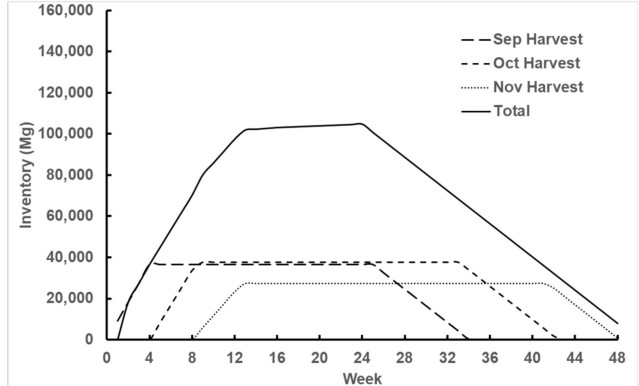

**Figure 6.** Inventory in SSLs for weeks 1 through 48 for the six-month window. (Results obtained with the Scenario 2 MATLAB program).

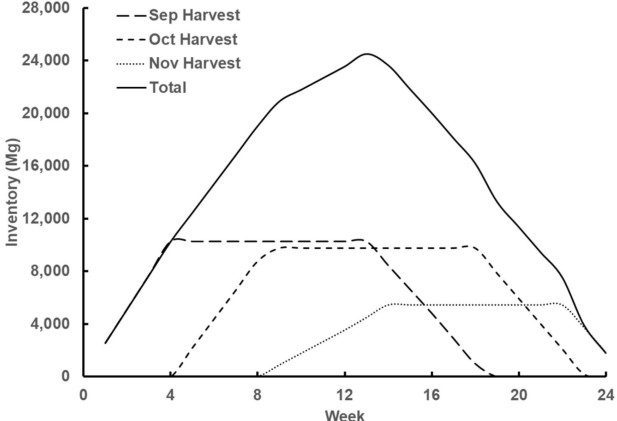

**Figure 7.** Inventory in SSLs for weeks 1 through 24 for the six-month window, six-month campaign. (Results obtained with the Scenario 3 MATLAB program).

**Table 5.** (**a**) Losses (% biorefinery design capacity) for Scenario 1. (**b**) Losses (% biorefinery design capacity) for Scenario 2. (**c**) Losses (% biorefinery design capacity) for Scenario 3.

| (a) | | | |
|---|---|---|---|
| **Loss Curves** | **Harvest Losses** | **Storage Losses** | **Total** |
| Low | 5.15 | 2.56 | 7.71 |
| High | 9.62 | 4.76 | 14.38 |
| (b) | | | |
| **Loss Curves** | **Harvest Losses** | **Storage Losses** | **Total** |
| Low | 8.48 | 2.22 | 10.70 |
| High | 16.89 | 3.86 | 20.75 |
| (c) | | | |
| **Loss Curves** | **Harvest Losses** | **Storage Losses** | **Total** |
| Low | 8.48 | 0.35 | 8.83 |
| High | 16.89 | 0.81 | 17.70 |

It is interesting that the total losses (harvest + storage) are greater for Scenario 2. This occurs because the harvest losses are so high in December, January, and February. Loss occurs when the biomass is stored standing in the field, and when it is stored in SSLs. The high harvest loss during the winter months dominate the comparison. These results, standing alone, do not make a strong argument for Scenario 2 compared to Scenario 1. The total losses for Scenario 3 fall between the other two options.

*5.4. Cost of Losses*

Calculated cost of losses incurred with Scenario 1 are given in Table 6a, in Table 6b for Scenario 2, and in Table 6c for Scenario 3. The same trends seen in Table 5a,b are observed. The cost of total losses is considerably higher for the six-month as compared to the three-month window. Total cost of losses is reduced 31% (low-loss curve) and 37% (high-loss curve) with a three-month window.

**Table 6.** (**a**) Cost of losses (USD/Mg biorefinery design capacity) for Scenario 1. (**b**) Cost of losses (USD/Mg biorefinery design capacity) for Scenario 2. (**c**) Cost of losses (USD/Mg biorefinery design capacity) for Scenario 3.

| (a) | | | |
|---|---|---|---|
| **Loss Curves** | **Harvest Losses** | **Storage Losses** | **Total** |
| Low | 4.37 | 2.12 | 6.49 |
| High | 8.96 | 3.92 | 12.88 |
| (b) | | | |
| **Loss Curves** | **Harvest Losses** | **Storage Losses** | **Total** |
| Low | 7.63 | 1.82 | 9.45 |
| High | 18.92 | 3.37 | 22.29 |
| (c) | | | |
| **Loss Curves** | **Harvest Losses** | **Storage Losses** | **Total** |
| Low | 7.63 | 0.28 | 7.91 |
| High | 18.92 | 0.65 | 19.57 |

*5.5. Fertilizer Cost*

Fertilizer cost factors for the two site conditions are given in Table 7. These data use the low harvest loss curve. The low field productivity condition, as well as the subsequent fertilizer cost,

is used in the subsequent comparison of the three scenarios. The USD/Mg fertilizer cost for Scenarios 2 and 3 are equal because the same area (expressed as ha/Mg biorefinery design capacity) is harvested each harvest month for both options.

**Table 7.** Fertilizer costs (USD/Mg biorefinery design capacity) for two field productivity levels.

| Field Productivity | 3-Month (Scenario 1) | 6-Month (Scenario 2) | 6-Month (Scenario 3) |
|---|---|---|---|
| Low | 14.26 | 12.95 | 12.95 |
| High | 19.39 | 17.77 | 17.77 |

### 5.6. Baler Cost

The baling cost corresponding to 175 h in Scenario 1 and 240 h in Scenarios 2 and 3 is repeated here.

Scenario 1 = 2.51 USD/Mg;
Scenarios 2 and 3 = 2.24 USD/Mg.

### 5.7. Comparison of Total Cost for Three Scenarios

For the comparison given here, the following costs are used, the "low" fertilizer cost, the "low" harvest loss curve, and the "high" storage loss curve. These curves were chosen because this performance likely can be achieved in commercial practice. Switchgrass standing in the field at 15% moisture content can be harvested with a mower-conditioner followed directly by the baler. However, adding a raking step to sweep more material into the window before harvest during the winter months will improve the collection of material that has fallen to the ground. This may have the negative consequence of introducing soil contaminants into the feedstock, in turn lowering feedstock quality. However, feedstock quality-based analysis of costs for the biorefinery and values to the producer is beyond the scope of this analysis.

The total cost figures in Table 8 are for comparison of the three scenarios; they are not a calculated biomass payment. It is interesting that the difference in Scenarios 1 and 2 is quite small, $30.33 - 29.71 = 0.62$ USD/Mg. The difference in losses cost is $8.29 - 11.00 = -2.71$ USD/Mg, which more than offsets the difference in storage cost $5.27 - 3.52 = 1.75$ USD/Mg. The expected advantage of Scenario 2, due to the smaller SSL storage capacity required, was erased when the cost of losses was considered.

**Table 8.** Difference in total cost (SSL storage + losses + fertilizer + baling) for three scenarios.

| Scenario 1 | | | | | | |
|---|---|---|---|---|---|---|
| Total Cost (USD/Mg) | 5.27 | + 8.29 | + 14.26 | + 2.51 | = | 30.33 |
| % of Total Cost | 17 | + 28 | + 47 | + 8 | = | 100% |
| **Scenario 2** | | | | | | |
| Total Cost (USD/Mg) | 3.52 | + 11.00 | + 12.95 | + 2.24 | = | 29.71 |
| % of Total Cost | 11 | + 37 | + 44 | + 8 | = | 100% |
| **Scenario 3** | | | | | | |
| Total Cost (USD/Mg) | 1.64 | + 8.28 | + 12.95 | + 2.24 | = | 25.11 |
| % of Total Cost | 6 | + 33 | + 52 | + 9 | = | 100% |

For a biorefinery processing 1 bale/min = 193,536 Mg/y, the cost difference in losses is about 0.5 million USD. Using Scenario 3 as a base, the total cost for Scenario 1 is about 21% higher, and it is 18% higher for Scenario 2.

## 6. Discussion

### 6.1. Comparison of Feedstock Contract Payments—Biorefinery

The average cost to the biorefinery (Mg/achieved annual capacity), considering the fertilizer and harvest loss adjustments, plus the cost of the storage losses (high curve), is given in Tables 9 and 10 for the low and high field productivity/fertilizer requirement, respectively. (It is re-emphasized, that the costs in these tables are averages for all biomass delivered for annual operation of the biorefinery). In general, the Scenario 2 cost is about 12% higher than Scenario 1 for the high harvest losses. The costs for the two scenarios are approximately equal when harvest losses are low. A biorefinery could choose Scenario 1 or 2 and expect minimal difference in average cost of feedstock.

**Table 9.** Average cost (USD/Mg) of feedstock purchased at the SSL for achieved annual biorefinery operation (low productivity/fertilizer requirement).

| Harvest Loss | 3-Month (Scenario 1) | 6-Month (Scenario 2) | 6-Month (Scenario 3) |
|---|---|---|---|
| Low | 93.79 | 93.32 | 87.92 |
| High | 98.95 | 104.93 | 99.00 |

**Table 10.** Average cost (USD/Mg) of feedstock purchased at the SSL for achieved annual biorefinery operation (high productivity/fertilizer requirement).

| Harvest Loss | 3-Month (Scenario 1) | 6-Month (Scenario 2) | 6-Month (Scenario 3) |
|---|---|---|---|
| Low | 94.66 | 93.49 | 88.09 |
| High | 99.82 | 104.95 | 99.02 |

### 6.2. Comparison of Feedstock Contract Payments—Feedstock Contractors

The data in this section are presented from the perspective of the feedstock contract holder. What would they be paid for the biomass they harvest to fulfill, for example, a September delivery contract, or a February delivery contract? This question is answered by applying the various adjustments defined in Equations (9)–(12).

Feedstock contract payments for direct delivery given in Table 11 are based on low fertilizer inputs (Figure 2) and low harvest losses (Figure 3). Obviously, for direct delivery, there is no storage loss adjustment. Potential feedstock contractors can examine the potential USD/ha income and compare with the gross income from their other enterprises. The USD/Mg results can be compared to the biorefinery average payment of 93.32 USD/Mg (Scenario 2) for the total biomass delivered for annual operation (Table 9). The cost in Table 9 is the average across the entire annual operation of the biorefinery, as compared to a payment to an individual feedstock contractor with a given month's contract. Moreover, it includes a storage loss adjustment.

**Table 11.** Feedstock contract payments for biomass placed in an SSL each month, no storage loss adjustment (Scenario 2). (Results obtained with Scenario 2 MATLAB program).

| Harvest Month | Feedstock Contract (USD/ha) | Feedstock Contract (USD/Mg) |
|---|---|---|
| 1 (September) | 566.81 | 84.30 |
| 2 (October) | 540.61 | 85.54 |
| 3 (November) | 518.91 | 86.72 |
| 4 (December) | 506.51 | 88.63 |
| 5 (January) | 496.11 | 89.98 |
| 6 (February) | 487.21 | 90.58 |

An *indication* of the influence of storage loss on the payment to an individual feedstock contractor is given in Table 12. Here, the cost of storage losses is given as USD/Mg of biomass placed in storage, rather than based on annual biorefinery design capacity. Thus, the value does not compare to the value given in Table 6b. This cost is hereafter referred to as a "storage loss adjustment". As previously explained, it was calculated by summing the total storage losses for the $i$th month harvest, multiplying by the value ($V_b + V_{HLi}$), and dividing by the total biomass placed in storage that month ($i$th month). When feedstock units from a contractor's SSL are delivered to the biorefinery, the mass arriving at the biorefinery (Mg) times the total payment in Table 12 is the total payment to the feedstock contractor.

**Table 12.** Feedstock contract payments showing influence of the storage loss adjustment (Scenario 2). (Results obtained with Scenario 2 MATLAB program).

| Harvest | Storage Loss Adjustment | Feedstock Contract | Total Payment |
|---|---|---|---|
| Month | Equation (11) (USD/Mg) | (USD/Mg at SSL) | (USD/Mg at Biorefinery) |
| 1 (September) | 5.44 | 84.30 | 89.74 |
| 2 (October) | 6.55 | 85.54 | 92.08 |
| 3 (November) | 7.26 | 86.72 | 93.98 |
| 4 (December) | 7.58 | 88.63 | 96.21 |
| 5 (January) | 7.25 | 89.98 | 97.23 |
| 6 (February) | 6.61 | 90.58 | 97.19 |

The cost of storage loss adjustment in Table 12 is a "weighted average" across all deliveries for the given harvest month. If an adjustment was calculated for each unit arriving at the biorefinery, it would range higher and lower around this average. For example, suppose a feedstock contractor has a September-delivery contract, and the biomass is shipped week 25, after the last direct shipment of February-harvest biomass. In this analysis, the second week of the month is the reference for a given harvest month, thus the number of weeks in storage is 25 − 2 = 23 weeks, and the storage loss adjustment for 23 weeks is applied. Another September-harvest contractor may have biomass shipped week 30, thus the number of weeks in storage is 30 − 2 = 28 weeks, and the storage loss adjustment for 28 weeks is applied. A contract based on calculations for each individual unit is not practical.

The payments in Table 12 were calculated to determine what each feedstock contract holder would be paid to ensure that they had an equal profit opportunity, whether they hold a September or February delivery contract. It is in the biorefinery's interest to write "distribution of delivery" contracts that provide the lowest average feedstock cost for annual operation. Consequently, the appropriate adjustments must be negotiated and mutually agreed to by both parties. The contract relationship between the feedstock contract holders and the biorefinery must be a win-win. Either all parties win, or all parties lose.

The decision on whether a biorefinery is built in a given rural community will depend on the leadership in that community and some level of shared risk. No investor will put their capital at risk to build a biorefinery until there are contracts in place for a reliable feedstock supply. No farmer will plant switchgrass unless they have a known market for biomass that ensures a profit.

## 7. Conclusions

The feedstock contract offered by a biorefinery in the Piedmont must ensure an equal profit opportunity for each contract holder. The analysis here envisions a business plan where the hauling and weekly delivery from SSLs is scheduled and paid for by the biorefinery. Storage time for an individual contractor's biomass is set by the biorefinery hauling schedule.

Costs for a three-month (September–November) and a six-month (September–February) harvest window to supply herbaceous biomass (switchgrass) to a biorefinery were compared. After applying an adjustment for fertilizer application, harvest loss differences based on harvest month, and storage losses based on number of weeks of storage in the SSL, the baseline feedstock

cost of 77 USD/Mg for 12-month operation increases to 93.79 and 93.32 USD/Mg for the three-month and six-month harvest windows, respectively. This is the average payment by the biorefinery for the total biomass delivered for annual operation. For comparison, the average payment for a six-month window to supply biomass for six months of operation was 87.92 USD/Mg.

Our specific study of the six-month window showed that, after applying a fertilizer adjustment and a harvest loss adjustment for harvest months September–February, payments to an individual contractor for direct harvest and delivery will range from 84.30 USD/Mg (September contract) to 90.58 USD/Mg (February contract). For stored biomass, an adjustment for storage loss in SSLs is applied, and the payments range from 89.74 to 97.19 USD/Mg.

**Supplementary Materials:** The following MATLAB program code are available online at http://www.mdpi.com/2624-7402/2/4/41/s1.

**Author Contributions:** Conceptualization, J.C., R.B.G., and J.F.; methodology, J.C. and R.B.G.; software, J.C.; validation, J.C. and R.B.G.; formal analysis, J.C.; investigation, J.C. and J.F.; data curation, J.C. and J.F.; writing—original draft preparation, J.C.; writing—review and editing, J.C., R.B.G., and J.F.; visualization, J.C.; supervision, R.B.G.; project administration, R.B.G.. All authors have read and agreed to the published version of the manuscript.

**Funding:** This research received no external funding

**Conflicts of Interest:** The authors declare no conflict of interest.

## Appendix A. Annual Cost to Build and Maintain SSL

The estimated cost to build an SSL was calculated by using data and procedures [60]. The site size for a 123 ha contract with average yield of 6.7 Mg/ha is used for the sample calculation.

$$(123 \text{ ha } (6.7 \text{ Mg/ha}))/(0.4 \text{ Mg/bale}) = 2060 \text{ round bales} \tag{A1}$$

Required area for round bales is as follows:

$$5 \text{ ft} \times 4 \text{ ft} \times (1 + 0.03) = 20.6 \frac{\text{ft}^2}{\text{bale}} \times (2060 \text{ bales}) = 42,436 \text{ ft}^2 = 3942 \text{ m}^2 \tag{A2}$$

RSMeans Data [60] for the relative size of 5000 yd$^2$ or 4180 m$^2$ site are used. For Rough Grading (RSMeans Data Code 31 22 13 with 40,000–45,000 S.F. (5000 S.Y.), p. 284), the labor is estimated at 960 USD, the equipment cost is estimated at 820 USD, and with overhead and profit estimates, costs total of 2350 USD.

The Aggregate Base Coarse (RSMeans Data Code 32 11 23, p. 362) assumes a layer that is 15 cm thick and a gravel size of 20 mm. For this pad size, estimates were as follows: materials, 5.35 USD/yd$^2$; the labor for installation, 0.41 USD/yd$^2$; equipment, 0.78 USD/yd$^2$; and including overhead and profit estimates, the total is 7.30 USD/yd$^2$. The total investment for aggregate base is as follows:

$$\frac{7.30 \text{ USD}}{\text{yd}^2} \left( 5000 \text{ yd}^2 \right) = 36,500 \text{ USD} \tag{A3}$$

According to the RSMeans Data [60], the Location Factor for Lynchburg, VA, is 87.8; thus, the estimated construction cost is as follows:

$$\frac{(2350 + 36,500)}{100} (87.8) = 34,110 \text{ USD} \tag{A4}$$

The construction cost is as follows:

$$\frac{34,110 \text{ USD}}{4180 \text{ m}^2} = 8.16 \text{ USD/m}^2 \tag{A5}$$

This estimate does not include general contractor markups on subcontractor work, general contractor office overhead and profit, and contingency. It also does not include a cost for erosion-control measures.

The annual cost to build and maintain SSL is calculated as follows:

Construction Cost: 34,110 USD;

Design life: 10 y, n =10;

Interest rate: 6.25%, r = 0.0625;

Insurance rate: $0.80/$100 value/y.

$$Ins = \frac{34,110}{100} (0.80) = 273 \text{ USD/y} \tag{A6}$$

Tax rate: 1%

$$Tax = 0.01 (34,110) = 341 \text{ USD/y} \tag{A7}$$

Repair and maintenance factor (R/M): 25% of construction cost over design life

$$0.25 \times 34{,}110/10 = 853 \text{ USD/y} \tag{A8}$$

Cost Recovery Factor:

$$CRF = \frac{r (1+r)^n}{(1+r)^n - 1} \tag{A9}$$

where r = interest rate (dec) and n = design life (y).

$$CRF = \frac{0.0625 (1 + 0.0625)^{10}}{(1 + 0.0625)^{10} - 1} = 0.137 \tag{A9a}$$

The annual cost is the sum of construction cost x cost recovery factor, plus insurance, taxes, and repair and maintenance:

$$34{,}110 (0.137) + 273 + 341 + 853 = 6140 \; USD \tag{A10}$$

The annual cost per square meter is as follows:

$$\frac{6140 \text{ USD}}{4180 \text{ m}^2} = 1.47 \text{ USD/m}^2 \tag{A11}$$

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
