# Peer review of "Feedstock Contract Considerations for a Piedmont Biorefinery"

_agriengineering, doi:10.3390/agriengineering2040041_

Round 1

Reviewer 1 Report

The Authors should improve their work according to the following indications.

1. In the introduction, the Authors should explain how the article has been structured by presenting the different sections.

2. Literature review is absent, and should be addressed. Moreover, the Authors should explain how their work fill the gap in the literature.

3. The Authors should discuss how the results can be interpreted in perspective of previous studies and of the working hypotheses.

4. Policy implications, limitations of the study and future research directions should be addressed.

5. Tables and figures should report the sources.

6. Extensive editing of English language and style required.

Reviewer 2 Report

The issue addressed in the paper discusses the feedstock contract considerations for a Piedmont biorefinery. Generally, this paper was prepared correctly, compatible with the scope of the journal. However, I recommend a few corrections to improve the quality of this article:
- to precisely define the research scenario (it is partially unclear and disorderly); needed to clarify the scope of the study and consequently a clear, step-by-step, simple, synthetic test pattern (see point 2); admittedly the methodology is described, but I recommend more precision, as the reader should know how to repeat a similar analysis on this basis;
- to clarify and supplement the computation procedure (see 2.3.)
- to explain briefly whether there is need to use, for instance, other methods,
that is, supplement the summary descriptive analysis.
I also strongly suggest that recommendations for specific, practical, not only general (and not entirely clear) applications of this research shall be provided (please complete point 5).
The language of this paper is relatively correct, however some descriptions would benefit from being more concise.

Round 2

Reviewer 1 Report

The Authors tried to improve their manuscript